# Is Prone Position [^18^F]FDG PET/CT Useful in Reducing Respiratory Motion Artifacts in Evaluating Hepatic Lesions?

**DOI:** 10.3390/diagnostics13152539

**Published:** 2023-07-31

**Authors:** Chung Won Lee, Hye Joo Son, Ji Young Woo, Suk Hyun Lee

**Affiliations:** 1Department of Radiology, Hallym University Kangnam Sacred Heart Hospital, Hallym University College of Medicine, Seoul 07441, Republic of Korea; cnddnjs1482@naver.com; 2Department of Nuclear Medicine, Dankook University Medical Center, Cheonan 31116, Republic of Korea; neuroscience@dankook.ac.kr

**Keywords:** fluorodeoxyglucose F18, positron emission tomography, prone position, liver

## Abstract

Prone position is useful in reducing respiratory motion artifacts in lung nodules on 2-Deoxy-2-[^18^F] fluoro-D-glucose ([^18^F]FDG) positron emission tomography/computed tomography (PET/CT). However, whether prone position PET/CT is useful in evaluating hepatic lesions is unknown. Thirty-five hepatic lesions from 20 consecutive patients were evaluated. The maximum standardized uptake value (SUV_max_) and metabolic tumor volume (MTV) of both standard supine position PET/CT and additional prone position PET/CT were evaluated. No significant difference in SUV_max_ (4.41 ± 2.0 vs. 4.23 ± 1.83; *p* = 0.240) and MTV (5.83 ± 6.69 vs. 5.95 ± 6.24; *p* = 0.672) was observed between supine position PET/CT and prone position PET/CT. However, SUV_max_ changes in prone position PET/CT varied compared with those in supine position PET/CT (median, −4%; range: −30–71%). Prone position PET/CT was helpful when [^18^F]FDG uptake of the hepatic lesions was located outside the liver on supine position PET/CT (*n* = 4, SUV_max_ change: median 15%; range: 7–71%) and there was more severe blurring on supine position PET/CT (*n* = 6, SUV_max_ change: median 11%; range: −3–32%). Unlike in lung nodules, prone position PET/CT is not always useful in evaluating hepatic lesions, but it may be helpful in individual cases such as hepatic dome lesions.

## 1. Introduction

Positron emission tomography (PET) is an imaging technique with high accuracy in diagnosing tumors and monitoring the effects of treatment [1]. However, PET does not provide anatomical information; therefore, other complementary imaging techniques, such as computed tomography (CT) are required. Consequently, several methods have been developed to register and merge PET and CT data, and a dual-modality device that integrates a radionuclide detector and a CT scanner in the same gantry for a single patient is currently being utilized in clinical practice [2,3]. The development of dual-modality PET/CT has enhanced the localization of lesions and enabled more precise diagnosis [4]. 2-Deoxy-2-[^18^F]fluoro-d-glucose ([^18^F]FDG) PET/CT is useful for detecting malignant hepatic lesions [5,6].

Attenuation correction has been demonstrated to enhance the quality of patient images, improve the detection of lesions, facilitate tumor staging, and enable better treatment monitoring in clinical settings compared to imaging without attenuation correction [7]. These effects can be further enhanced through CT-based attenuation, offering the advantages of reduced statistical noise, faster acquisition times, and improved patient comfort [8,9].

During the fusion of PET and CT, artifacts may occur due to respiratory motion, metallic implants, CT contrast media, and truncation. Among them, respiratory motion artifacts are one of the most common and important artifacts in PET/CT [10,11]. Moreover, respiration causes the misregistration of liver locations on PET and CT, which leads to inaccurate attenuation corrections [12,13,14,15,16,17,18]. The gating method has been attempted to reduce artifacts caused by respiratory motion on PET/CT, using an external sensor, such as a pressure belt or a video camera, or data-driven respiratory gating [19,20,21]. However, there is a limitation to the use of respiratory gating methods because an additional external device is required for gating, or some PET/CT devices do not support such methods. In contrast, [^18^F]FDG PET/CT imaging in the prone position is an effective method for reducing respiratory motion artifacts in lung nodules without an external device or software for data-driven respiratory gating [22].

In the prone position, the spontaneous effort of breathing decreases [23]. Moreover, since the movement of the diaphragm in the prone position is reduced [24,25,26], prone position PET/CT (pPET/CT) might also be helpful in evaluating hepatic lesions located close to the diaphragm. However, pPET/CT has multiple disadvantages; for example, it requires additional radiation exposure and extends the image acquisition time. Despite these drawbacks, no studies have investigated whether pPET/CT can aid in the assessment of hepatic lesions by reducing respiratory motion artifacts. Therefore, this study was designed to determine whether pPET/CT is helpful in evaluating hepatic lesions and to compare pPET/CT with standard supine position PET/CT (sPET/CT).

## 2. Materials and Methods

### 2.1. Subjects

Patients who underwent [^18^F]FDG PET/CT to evaluate hepatic lesions in our hospital from January 2021 to December 2021 were enrolled. For patients with hepatic lesions on CT or MRI, additional imaging was performed in the prone position (Figure 1), depending on the decision made by the nuclear medicine board-certified physician (SHL). Among these hepatic lesions, those with a size larger than 3 cm or a [^18^F]FDG uptake unrelated to a true hepatic lesion, such as percutaneous transhepatic biliary drainage catheter insertion, were excluded from the analysis (Figure 2). The patients’ age, sex, weight, reason for PET/CT, number of measured hepatic lesions, diagnosis and confirmation of hepatic lesions, and location of hepatic lesions were recorded based on the electronic medical record and images uploaded to the picture archiving and communication system. This study was conducted in accordance with the Declaration of Helsinki (as revised in 2013) and approved by the Institutional Review Board of our institution (IRB no. 2022-05-039). Written informed consent was not needed because this study was retrospective and all the participants’ data were anonymized.

### 2.2. PET/CT Imaging Protocol

According to the recent guidelines for tumor imaging [27], we acquired the images under the following conditions: before [^18^F]FDG PET/CT imaging, the patient fasted for at least 6 h, and the blood glucose level of the patient was measured; the patient was administered 5.18-MBq/kg (0.14 mCi/kg) [^18^F]FDG; and the blood glucose level was controlled so that it remained <8.33 mmol/L (150 mg/dL). Image acquisition was initiated within 50–70 min after [^18^F]FDG injection using a Gemini TF 16 PET/CT scanner (Philips Healthcare, Cleveland, OH, USA). After the initial low-dose CT study (120 kVp, 50 mAs), PET images were acquired in the 3D mode from the base of the skull to the mid-thigh with 7–10 beds of 2 min each. The PET images were reconstructed using the 3D RAMLA iterative OSEM algorithm (3 iterations, 33 subsets, no filtering) with a CT-based attenuation correction.

Immediately after the completion of the standard torso sPET/CT, the patient was placed in the prone position, and additional abdominal pPET/CT was performed. The field covered the upper and lower margins of the liver. The image acquisition setting and reconstruction algorithm of pPET/CT were the same as those of sPET/CT.

### 2.3. PET/CT Image Analysis

Various parameters of sPET/CT and pPET/CT were measured on a workstation (Advantage Workstation 4.7, GE Healthcare, Milwaukee, WI, USA) by an experienced nuclear medicine board-certified physician (SHL). We measured the nodule’s maximum standardized uptake value (SUV_max_) and metabolic tumor volume (MTV), with a threshold of 50% within a container volume of interest around the hepatic lesion. In each sPET/CT and pPET/CT image, the location of hepatic lesions was recorded among the anterior, middle, and posterior regions. Additionally, we measured the difference in diaphragm position between PET and CT (DDP) images, which was defined as the difference measured in the vertical direction distance of the hepatic dome upper margin level of non-attenuation correction PET and CT images in the fusion image (Figure 3). To observe the upper margin of the liver during measurement, similar to the method used by Van der Vos et al. [28], we set the CT image to a liver window setting (width, 175; level, 45) and arbitrarily adjusted the window level and width of the non-attenuation correction PET image. After the measurement, 2 nuclear medicine specialists (SHL and HJS), who were blinded to the patient’s clinical information, categorized the possible reasons for SUV_max_ changes of hepatic lesions on pPET/CT compared with those on sPET/CT by visual inspection. If the results were different, the 2 interpreters reviewed them together and reached a consensus.

### 2.4. Statistical Analysis

The paired *t*-test was used to compare SUVmax, MTV, and DDP between sPET/CT and pPET/CT. The Wilcoxon test was used to compare the location of hepatic lesions between sPET/CT and pPET/CT. *p*-values of less than 0.05 were used to denote statistical significance. Statistical analyses were performed using the Statistical Package for the Social Sciences, version 27 (IBM Corp., Armonk, NY, USA).

## 3. Results

Among the 20 patients, we analyzed 35 hepatic lesions (Figure 2). Of the 35 hepatic lesions, 33 (94%) were malignant and 2 (6%) were hepatic abscesses. The lesions were diagnosed either pathologically or clinically. The characteristics of the patients and hepatic lesions are summarized in Table 1 and Table 2.

Hepatic lesions tended to be located more anteriorly on pPET/CT than on sPET/CT. No significant differences in the SUV_max_ and MTV of the hepatic lesions were observed between sPET/CT and pPET/CT (*p* = 0.240 and 0.672, respectively, Table 3). However, SUV_max_ changes of hepatic lesions on pPET/CT from sPET/CT varied from 71% to −30% (median = −4%) (Figure 4). The possible reasons for the SUV_max_ changes were categorized into five, as shown in Table 4. Ten hepatic lesions (29%) showed reduced artifacts on pPET/CT compared with those on sPET/CT (four hepatic lesions on PET were located outside the liver on CT in sPET/CT and six hepatic lesions showed more blurring on sPET/CT), thirteen hepatic lesions (37%) showed more severe artifacts on pPET/CT than on sPET/CT (twelve hepatic lesions showed more blurring on pPET/CT and one hepatic lesion on PET was located outside the liver on CT on pPET/CT), and the remaining twelve lesions (34%) showed no significant difference in artifacts between pPET/CT and sPET/CT. Among them, the reason for the largest increase in the SUV_max_ on pPET/CT was when the hepatic lesion on PET was located outside the liver on CT in sPET/CT (Figure 5). pPET/CT showed less diaphragm position difference between PET and CT than sPET/CT (5.1 ± 6.9 mm vs. 11.4 ± 6.8 mm; *p* = 0.009).

## 4. Discussion

[^18^F]FDG PET/CT is useful in detecting hepatic metastasis. According to D’Souza et al., the sensitivity of this imaging modality is 97% and its specificity is 75% [29]. Moreover, [^18^F]FDG PET/CT has also been reported to be sensitive to primary malignant tumors, such as intrahepatic cholangiocarcinoma [30,31,32]. Additionally, there is an increase in the [^18^F]FDG uptake in non-malignant lesions, such as hepatic abscesses [33]. In this study, most hepatic lesions with increased [^18^F]FDG uptake were metastases (*n* = 30, 86%), intrahepatic cholangiocarcinomas (*n* = 3, 9%), and abscesses (*n* = 2, 6%) (Table 1.)

However, respiratory motion causes major artifacts on [^18^F]FDG PET/CT images for pulmonary and hepatic lesions. Because of the long acquisition time of PET scans, they are acquired when the patient is breathing freely [34]. Thus, the final image is a result of the average of multiple breathing cycles. Respiratory motion artifacts result from discrepancies in the anatomy of the thoracic and abdominal organs on CT images, as well as averaging over many respiratory cycles during PET studies [34]. As described by Papathanassiou et al. [16], this phenomenon sometimes causes misregistration of lesions between the two modalities or interferes with image fusion of normal organs, as well as causing erroneous attenuation correction. Because of respiratory motion, the density of a particular organ can be attributed to areas with different densities. This can also cause misregistration of hepatic lesions near the diaphragm and SUV_max_ errors [11,13]. Moreover, the movement of the diaphragm can result in misalignment between CT and PET images, particularly in the lower lung and upper abdomen regions, and in some cases, the misregistration can be significant. This misalignment can cause false appearance of hepatic lesions that mimic pulmonary nodules at the base of the lungs [35]. Moreover, the imprecise localization of photons caused by diaphragmatic motion and suboptimal attenuation correction maps can lead to inaccurate SUVs for lesions located near the diaphragm in PET attenuation-corrected images [36].

Several studies have been conducted to address and minimize respiratory motion artifacts. Among these studies, there is evidence that the use of respiratory gating devices could improve detectability and quantification of lesions [37,38,39]. According to Pepin et al. [20], methods using several sensors for motion tracking in the gating method have been developed. First, a pressure sensor is coupled with an elastic chest belt to measure the pressure difference in the abdominal wall during the respiratory cycle. Second, an infrared reflective marker is placed on a plastic box located on the patient’s chest and recorded with a video camera to obtain a signal for processing [40]. A third method for measuring the temperature change during respiration has been developed. The method is based upon the principle that air warms as it passes through the lungs; hence, a high-sensitivity thermistor located inside a conventional oxygen mask is required. Another method is to place the probe in the patient’s nostril [41]. The fourth method uses a spirometer placed in the patient’s nostrils and inside the mouth to measure the inhalation and exhalation volumes [42]. The fifth method performs imaging in the prone position to decrease the spontaneous effort of breathing and reduce movement of the diaphragm, which has been proven to reduce the respiratory motion artifacts of lung nodules without an external device or software for data-driven respiratory gating [22].

The respiratory-gated list-mode PET data can be processed in two different manners (multi-bin and single-bin methods) [20]. Multi-bin methods include time-based and amplitude-based methods. First, in time-based processing, the PET data are divided into specific time intervals or bins corresponding to different phases of the respiratory cycle [43,44]. In amplitude-based processing, the PET data are divided according to the amplitude of respiratory motion [20]. Single-bin methods include the deep breathing method and a CT-based method [43,44]. However, these methods are not clinically convenient because they require an external device, additional software, post-processing, and patient effort for regular breathing. Recently, data-driven respiratory gating has been used [19,21], but this method is not applicable to conventional PET/CT, and it requires extra expenses. Therefore, there is a clinical need for simple methods to reduce respiratory motion artifacts. pPET/CT was recently reported as a simple method for reducing respiratory motion artifacts in evaluating lung nodules [22]. Therefore, we assumed that pPET/CT would aid in evaluating hepatic lesions. Although no statistically significant difference was observed, the change in the SUV_max_ varied from −30% to 71% between sPET/CT and pPET/CT, and pPET/CT was helpful in several individual cases.

To the best of our knowledge, this is the first attempt to utilize pPET/CT for the purpose of mitigating respiratory motion artifacts in the evaluation of hepatic lesions. The two main instances in which pPET/CT was helpful were (1) when hepatic lesions on PET were located outside the liver on CT in sPET/CT; and 2) when severe blurring was observed on sPET/CT (Table 4). In this study, four hepatic lesions (11%) were located in the lung on CT in sPET/CT, and these lesions showed a 15% increase in the median SUV_max_ in pPET/CT (Table 4). If the [^18^F]FDG uptake of the hepatic lesion on PET is located in the lung on CT, which is outside of the liver, SUV_max_ would be underestimated because significantly low attenuation of the lung compared with the hepatic lesion results in inaccurate attenuation correction [22]. Furthermore, the reason why the blurring in six hepatic lesions (17%) decreased on pPET/CT is thought to be that diaphragmatic respiration motion was decreased in the prone position [24,25,26]. Our result that the DDP of pPET/CT was significantly lower than that of sPET/CT supports the two aforementioned reasons.

Nevertheless, taking all hepatic lesions together, the mean SUV_max_ and MTV were not significantly different between sPET/CT and pPET/CT. We considered the following reasons why pPET/CT is less useful in evaluating hepatic lesions than in evaluating lung nodules.

First, unlike lung nodules, hepatic lesions tend to move forward in the prone position. According to the results of this study, six hepatic lesions (17%) moved from the middle region to the anterior region, and four hepatic lesions (11%) moved from the posterior region to the middle region (Table 3). In the study by Lee et al., pPET/CT improved vertical direction respiratory motion artifacts of most lung nodules. However, the misregistration of some lung nodules located in the anterior region tended to worsen in a horizontal direction [22]. According to Shin et al., in the prone position, the respiratory motion of the posterior lung is greatly decreased compared with that in the supine position, whereas the respiratory motion of the anterior lung is slightly increased [45]. Therefore, the location expected to have an advantage due to reduced respiratory movement in the prone position is mainly the posterior lesion; however, hepatic lesions appear to have little advantage from the prone position because they are located relatively anteriorly.

Second, there is a significant difference in attenuation between lung nodules and the background lung; however, there is little difference in attenuation between hepatic lesions and the background liver. If the [^18^F]FDG uptake of the lung nodule located outside the lung nodule on CT, SUV_max_ would be underestimated because a significantly low attenuation of the background lung results in inaccurate attenuation correction [22]. However, even if misregistration occurs in hepatic lesions, attenuation correction may not be significantly affected because the attenuation of hepatic lesions and the background liver is similar. Exceptionally, the prone position is also helpful when the hepatic lesion on PET is located in the lung, which is outside the liver, on CT.

The advantage of pPET/CT is that it can be performed without special equipment. Recent respiratory gating methods require external respiratory monitoring devices or software [46]. The disadvantages of pPET/CT are that patients are exposed to additional radiation and that this imaging modality takes longer to change the patient’s position.

This study has several limitations. First, pPET/CT only covered the abdomen, whereas sPET/CT covered the entire area from the skull base to the proximal thighs. Therefore, it is unclear whether the benefit of pPET/CT was from prone positioning or from the shorter time difference between the PET and CT of regional abdomen imaging compared torso imaging. Second, because pPET/CT was taken 20 min later than sPET/CT, the SUV_max_ values may have been affected by the time difference, not only by the position change. Third, because the respiratory gating method was not compared in this study, whether pPET/CT is a better method than the respiratory gating method in evaluating hepatic lesions is unknown. Because pPET/CT can be performed in combination with other respiratory gating methods, whether combining these methods presents an an advantage would be worth investigating in future studies.

## 5. Conclusions

There was no statistically significant difference in the SUV_max_ and MTV between sPET/CT and pPET/CT in evaluating hepatic lesions. However, the changes in the SUV_max_ for each hepatic lesion varied. Although not always, pPET/CT may be helpful in evaluating hepatic lesions, particularly when the [^18^F]FDG uptake of the hepatic lesion is located outside the liver on CT and when severe blurring due to respiratory motion is observed on sPET/CT.

## Figures and Tables

**Figure 1 diagnostics-13-02539-f001:**
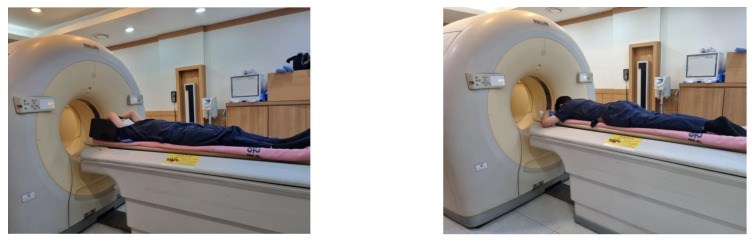
Image acquisition of PET/CT in the supine (**left**) and prone (**right**) positions.

**Figure 2 diagnostics-13-02539-f002:**
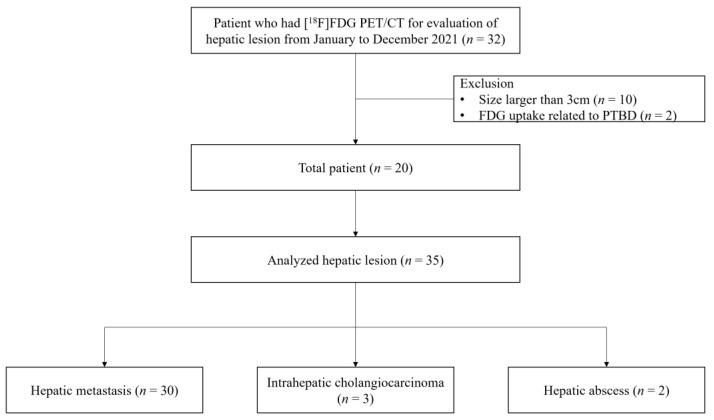
Flow diagram of patient enrollment.

**Figure 3 diagnostics-13-02539-f003:**
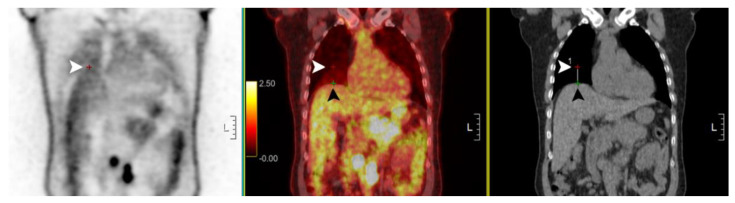
Measurement of diaphragm level difference between PET and CT. The upper margin of the liver on PET (white arrowhead) and CT (black arrowhead) was presumed as the diaphragm level and the level difference was measured on the fusion image.

**Figure 4 diagnostics-13-02539-f004:**
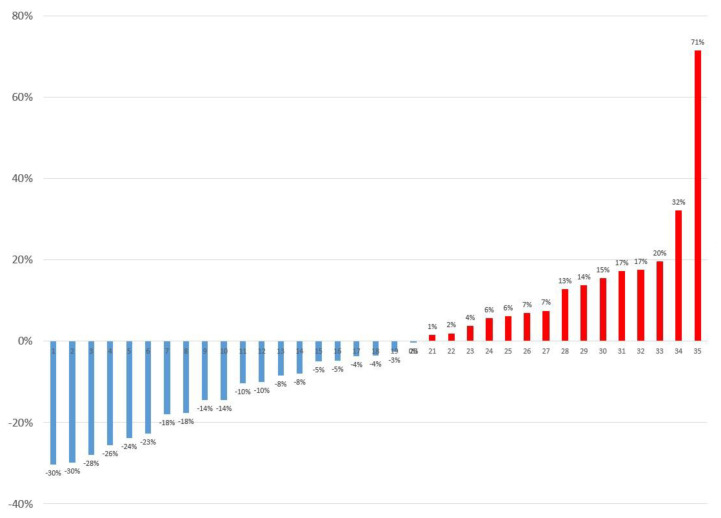
SUVmax change of hepatic lesions in prone position PET/CT compared to supine position PET/CT (*n* = 35).

**Figure 5 diagnostics-13-02539-f005:**
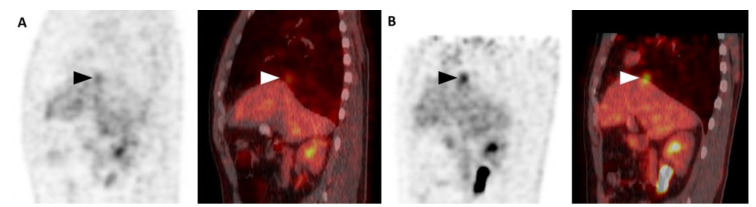
Representative case. The [^18^F]FDG uptake of the hepatic dome lesion (size: 13 mm, arrowheads) is located in the lung in the supine position PET/CT (**A**), but close to the liver in the prone position PET/CT (**B**). The SUV_max_ increased from 2.52 to 4.32 (71% increase).

**Table 1 diagnostics-13-02539-t001:** Patient characteristics.

Patient Characteristics	*n* = 20
Age at diagnosis, year (mean ± SD)	67.0 ± 9.7
Weight, kg (mean ± SD)	62.5 ± 9.0
Sex	
Male, *n* (%)	14 (70%)
Female, *n* (%)	6 (30%)
Reason for [^18^F]FDG PET/CT	
Diagnosis and initial staging, *n* (%)	15 (75%)
Recurrence, *n* (%)	5 (25%)
Number of measured hepatic lesions	
1	12 (60%)
2	5 (25%)
4	2 (10%)
5	1 (5%)

SD, standard deviation; [^18^F]FDG, 2-Deoxy-2-[^18^F]fluoro-D-glucose; PET, positron emission tomography; CT, computed tomography.

**Table 2 diagnostics-13-02539-t002:** Lesion characteristics.

Lesion Characteristics	*n* = 35
Size, mm (mean ± SD)	13.0 ± 5.8
Diagnosis of hepatic lesions	
Hepatic metastasis, *n* (%)	30 (86%)
Pancreas cancer, *n* (%)	8 (23%)
Breast cancer, *n* (%)	6 (17%)
Gastric cancer, *n* (%)	6 (17%)
Urothelial cancer, *n* (%)	4 (11%)
Colorectal cancer, *n* (%)	3 (9%)
Lung cancer, *n* (%)	1 (3%)
Common bile duct cancer, *n* (%)	1 (3%)
Subglottic cancer, *n* (%)	1 (3%)
Intrahepatic cholangiocarcinoma, *n* (%)	3 (9%)
Hepatic abscess, *n* (%)	2 (6%)
Confirmation of hepatic lesions	
Pathological, *n* (%)	6 (17%)
Clinical, *n* (%)	29 (83%)
Location	
I, *n* (%)	0 (0%)
II, *n* (%)	6 (17%)
III, *n* (%)	2 (6%)
IV, *n* (%)	8 (23%)
V, *n* (%)	3 (9%)
VI, *n* (%)	7 (20%)
VII, *n* (%)	3 (9%)
VIII, *n* (%)	6 (17%)
Distance from the diaphragm, mm (mean ± SD)	32.2 ± 25.1

SD, standard deviation.

**Table 3 diagnostics-13-02539-t003:** [^18^F]FDG PET/CT findings of hepatic lesions (*n* = 35).

Parameters	Supine	Prone	*p*-Value
SUV_max_, (mean ± SD)	4.41 ± 2.05	4.23 ± 1.83	0.240
MTV, cm^3^ (mean ± SD)	5.83 ± 6.69	5.95 ± 6.24	0.672
Location			
Anterior, *n* (%)	17 (49%)	23 (66%)	0.005 *
Middle, *n* (%)	10 (29%)	8 (23%)	
Posterior, *n* (%)	8 (23%)	4 (11%)	

[^18^F]FDG, 2-Deoxy-2-[^18^F]fluoro-D-glucose; PET, positron emission tomography; CT, computed tomography; SD, standard deviation; SUV_max_, maximum standardized uptake value; MTV, metabolic tumor volume. * *p* < 0.05 was considered statistically significant.

**Table 4 diagnostics-13-02539-t004:** Possible reasons for SUV_max_ changes by visual inspection (*n* = 35).

Reasons	Change of SUV_max_ (%)		*n* (%)
	Median	Range	
[^18^F]FDG uptake outside the liver on CT in sPET/CT	15%	[7% to 71%]	4 (11%)
More blurring in sPET/CT	11%	[−3% to 32%]	6 (17%)
Unremarkable	1%	[−8% to 18%]	12 (34%)
More blurring in pPET/CT	−19%	[−30% to −8%]	12 (34%)
[^18^F]FDG uptake outside the liver on CT in pPET/CT	−30%	[−30% to −30%]	1 (3%)
Total patients	−4%	[−30% to 71%]	35 (100%)

SUV_max_, maximum standardized uptake value; [^18^F]FDG, 2-Deoxy-2-[^18^F]fluoro-D-glucose; sPET/CT, supine position PET/CT; pPET/CT, prone position PET/CT; PET, positron emission tomography; CT, computed tomography.

## Data Availability

Not applicable.

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
