# Peer review of "Is Prone Position [18F]FDG PET/CT Useful in Reducing Respiratory Motion Artifacts in Evaluating Hepatic Lesions?"

_diagnostics, 2023, doi:10.3390/diagnostics13152539_

Round 1

Reviewer 1 Report

Comments:

The current work represents the possibility of using FDG-pPET/CT for better hepatic lesion detection vs supine position. As results, authors claim that there is no significant difference among prone vs supine position, except individual cases. The proposed idea is interesting and the final achievements performed on 20 real pts can be considered for publishing. However there are some points that should be clarified:

·         What about patient weight while implementing the proposed strategy at prone position vs supine position. It is better to indicate to pt weight at table including pt characteristics.

·         By increasing the number of patients, Does the obtained results and statistical analysis change? Please specify that.

·         Introduction section should be revise to highlight the novelties of the work.

·         Please add a last paragraph at Introduction section including local discussion on the results conceptually.

·         Before fig. 1, it is better to add another new schematic figure showing prone position of a patient typically.

·         It is suggested to renew of some old references used in this work.  

·         The English of the work is good but further revising can improve the texts.

Comments:

The current work represents the possibility of using FDG-pPET/CT for better hepatic lesion detection vs supine position. As results, authors claim that there is no significant difference among prone vs supine position, except individual cases. The proposed idea is interesting and the final achievements performed on 20 real pts can be considered for publishing. However there are some points that should be clarified:

·         What about patient weight while implementing the proposed strategy at prone position vs supine position. It is better to indicate to pt weight at table including pt characteristics.

·         By increasing the number of patients, Does the obtained results and statistical analysis change? Please specify that.

·         Introduction section should be revise to highlight the novelties of the work.

·         Please add a last paragraph at Introduction section including local discussion on the results conceptually.

·         Before fig. 1, it is better to add another new schematic figure showing prone position of a patient typically.

·         It is suggested to renew of some old references used in this work.  

·         The English of the work is good but further revising can improve the texts.

Author Response

We thank the reviewer for their insightful comments on our manuscript. We have made every effort to address your concerns and revised the manuscript according to your suggestions. Please read the attached word file for details.

__________________________________________________________________________________

Reviewer #1:

Q. What about patient weight while implementing the proposed strategy at prone position vs supine position. It is better to indicate to pt weight at table including pt characteristics.

A. Thank you for your insightful suggestion. We have included the mean patient weight in Table 1 as you advised.

Table 1. Patient characteristics

Patient characteristics

n = 20

Age at diagnosis, year (mean ± SD)

67.0 ± 9.7

Weight, kg (mean ± SD)

62.5 ± 9.0

Sex

Male, n (%)

14 (70%)

Female, n (%)

6 (30%)

Reason for [18F]FDG PET/CT

Diagnosis and initial staging, n (%)

15 (75%)

Recurrence, n (%)

5 (25%)

Number of measured hepatic lesions

1

12 (60%)

2

5 (25%)

4

2 (10%)

5

1 (5%)

SD, standard deviation; [18F]FDG , 2-Deoxy-2-[18F]fluoro-D-glucose; PET, positron emission tomography; CT, computed tomography.

Q. By increasing the number of patients, Does the obtained results and statistical analysis change? Please specify that.

A. Thank you for your comment. Having conducted PET/CT in the prone position for hepatic lesion evaluation for more than a year at our institution, we found that only a small number of patients benefit from this technique considering the additional radiation exposure and extended image acquisition time. Thereby, we have stopped performing additional prone position PET/CT for hepatic lesion evaluation, and thus, we could not increase the number of patients. Furthermore, we anticipate that increasing the number of patients would cause proportional increase in patients with better and worse outcomes with prone position PET/CT. Therefore, we strongly believe that the results would not become significant by increasing the patient population, which are currently insignificant.

Q. Introduction section should be revise to highlight the novelties of the work.

A. Thank you for addressing this issue. We have added the following sentence in the Introduction section to highlight the novelty of our work.

  • Page 2 line 62: Despite these drawbacks, no studies have investigated whether pPET/CT can aid in the assessment of hepatic lesions by reducing respiratory motion artifacts.

Q. Please add a last paragraph at Introduction section including local discussion on the results conceptually.

A. Thank you for your suggestion. Accordingly, we have modified the last paragraph in the Introduction section.

  • Page 2 line 58: In the prone position, the spontaneous effort of breathing decreases [23]. Moreover, since the movement of the diaphragm in the prone position is reduced [24–26], prone position PET/CT (pPET/CT) might also be helpful in evaluating hepatic lesions located close to the diaphragm. However, pPET/CT has multiple disadvantages; for example, it requires additional radiation exposure and extends the image acquisition time. Despite these drawbacks, no studies have investigated whether pPET/CT can aid in the assessment of hepatic lesions by reducing respiratory motion artifacts. Therefore, this study was designed to determine whether pPET/CT is helpful in evaluating hepatic lesions and to compare pPET/CT with standard supine position PET/CT (sPET/CT).

Q. Before fig. 1, it is better to add another new schematic figure showing prone position of a patient typically.

A. Thank you for this comment. We have added an additional figure as you suggested.

(Figures are included in the attached word file.)

Figure 1. Image acquisition of PET/CT in the supine (left) and prone (right) positions

Q. It is suggested to renew of some old references used in this work.

A. Thank you for your suggestion. Accordingly, we have replaced the following references using the existing content as a basis.

References

  1. Weber, W.A.; Avril, N.; Schwaiger, M. Relevance of positron emission tomography (PET) in oncology. Strahlenther Onkol 1999, 175, 356-373, doi:10.1007/s000660050022.

Moon, S.H.; Cho, Y.S.; Choi, J.Y. KSNM60 in Clinical Nuclear Oncology. Nucl Med Mol Imaging 2021, 55, 210-224, doi:10.1007/s13139-021-00711-9

  1. Delbeke, D.; Martin, W.H.; Sandler, M.P.; Chapman, W.C.; Wright, J.K., Jr.; Pinson, C.W. Evaluation of benign vs malignant hepatic lesions with positron emission tomography. Arch Surg 1998, 133, 510-515; discussion 515-516, doi:10.1001/archsurg.133.5.510.

Ozaki, K.; Harada, K.; Terayama, N.; Kosaka, N.; Kimura, H.; Gabata, T. FDG-PET/CT imaging findings of hepatic tumors and tumor-like lesions based on molecular background. Jpn J Radiol 2020, 38, 697-718, doi:10.1007/s11604-020-00961-1.

  1. Lonneux, M.; Borbath, I.; Bol, A.; Coppens, A.; Sibomana, M.; Bausart, R.; Defrise, M.; Pauwels, S.; Michel, C. Attenuation correction in whole-body FDG oncological studies: the role of statistical reconstruction. Eur J Nucl Med 1999, 26, 591-598, doi:10.1007/s002590050426.

Joshi, U.; Raijmakers, P.G.; Riphagen, II; Teule, G.J.; van Lingen, A.; Hoekstra, O.S. Attenuation-corrected vs. nonattenuation-corrected 2-deoxy-2-[F-18]fluoro-D-glucose-positron emission tomography in oncology: a systematic review. Mol Imaging Biol 2007, 9, 99-105, doi:10.1007/s11307-007-0076-5.

Q. The English of the work is good but further revising can improve the texts.

A. We appreciate your feedback. Accordingly, we have revised awkward phrasing and sentences to enhance readability and clarity.

Reviewer 2 Report

Positron emission tomography (PET) is an technique that shows high accuracy in diagnosing tumors and monitoring the effects of treatments. PET imaging does not provide anatomical information, requiring other complementary imaging techniques, such as computed tomography (CT) for anatomical description.

During PET and CT fusion, artifacts produced by respiratory movements, metal implants, CT contrast medium may appear.

pPET/CT can assess liver lesions, especially when FDG uptake of the liver lesion is located outside the liver as shown on CT and when the images cannot be highlighted due to respiratory movements on sPET/CT. The study uses pPET/CT in order to reduce respiratory motion artifacts in the evaluation of liver lesions.

pPET/CT proved to be useful when liver lesions were located outside liver, as shown on PET, or when a severe blurring was observed on sPET/CT.

There are some disadvantages for pPET/CT related to additional radiation exposure, a longer period of time for changing the patient, s  position, limited exploration of the abdomen while sPET/CT can cover the entire area from the base of the skull to the proximal thighs.

I consider that the information presented is useful for medical practice. I recommend it for publication. 

Author Response

We thank the reviewer for valuable comments.

Reviewer 3 Report

The authors analyze a topic which is of interest – if the prone position in PET/CT is useful in reducing respiratory motion artifacts in evaluating hepatic lesions. This is the first attempt to utilize PET/CT for the purpose of mitigating respiratory motion artifacts in the evaluation of hepatic lesions and continues researches of the authors on lungs.

The presentation is clear, comprehensive and well documented.

The references are appropriate, up-to-date and contain 46 titles.

The figures are appropriate and mandatory for sustaining the topic and contain illustrative imaging.

The 4 tables offer information on the patients and statistical data.

I found no plagiarism.

The discussions and conclusions are coherent and connected to the content.

It is a retrospective study and contains a small number of patients but brings new ideas.

In my opinion the paper fits the journal and the language is correct and understandable.

I recommend the paper to be accepted.

Author Response

We are grateful for your thoughtful review.